# Effects of the COVID-19 pandemic on the mental health of clinically extremely vulnerable children and children living with clinically extremely vulnerable people in Wales: a data linkage study

Laura Elizabeth Cowley [1], Karen Hodgson,[2] Jiao Song,[3] Tony Whiffen,[4] Jacinta Tan,[5] Ann John [1], Amrita Bandyopadhyay [6], Alisha R Davies [2]

For numbered affiliations see end of article.

**Correspondence to**
Dr Laura Elizabeth Cowley;
l.e.cowley@swansea.ac.uk

## ABSTRACT

**Objectives** To determine whether clinically extremely vulnerable (CEV) children or children living with a CEV person in Wales were at greater risk of presenting with anxiety or depression in primary or secondary care during the COVID-19 pandemic compared with children in the general population and to compare patterns of anxiety and depression during the pandemic (23 March 2020–31 January 2021, referred to as 2020/2021) and before the pandemic (23 March 2019–31 January 2020, referred to as 2019/2020), between CEV children and the general population.

**Design** Population-based cross-sectional cohort study using anonymised, linked, routinely collected health and administrative data held in the Secure Anonymised Information Linkage Databank. CEV individuals were identified using the COVID-19 shielded patient list.

**Setting** Primary and secondary healthcare settings covering 80% of the population of Wales.

**Participants** Children aged 2–17 in Wales: CEV (3769); living with a CEV person (20 033); or neither (415 009).

**Primary outcome measure** First record of anxiety or depression in primary or secondary healthcare in 2019/2020 and 2020/2021, identified using Read and International Classification of Diseases V.10 codes.

**Results** A Cox regression model adjusted for demographics and history of anxiety or depression revealed that only CEV children were at greater risk of presenting with anxiety or depression during the pandemic compared with the general population (HR=2.27, 95% CI=1.94 to 2.66, p<0.001). Compared with the general population, the risk among CEV children was higher in 2020/2021 (risk ratio 3.04) compared with 2019/2020 (risk ratio 1.90). In 2020/2021, the period prevalence of anxiety or depression increased slightly among CEV children, but declined among the general population.

**Conclusions** Differences in the period prevalence of recorded anxiety or depression in healthcare between CEV children and the general population were largely driven by a reduction in presentations to healthcare services by children in the general population during the pandemic.

## STRENGTHS AND LIMITATIONS OF THIS STUDY

⇒ Strengths of this study include its national focus and clinical relevance; to our knowledge, this is the first population-based study examining the effects of the COVID-19 pandemic on healthcare use for anxiety or depression among clinically extremely vulnerable (CEV) children and children living with a CEV person in Wales.

⇒ We compared 2020/2021 data with prepandemic 2019/2020 data for CEV children and children in the general population, to place the impact of the COVID-19 pandemic in the context of longer-term patterns of healthcare use.

⇒ We used a novel approach and linked multiple datasets to identify a cohort of children living with a CEV person in Wales during the COVID-19 pandemic.

⇒ There was heterogeneity within the shielded patient list that was used to create the cohorts of children identified as CEV or living with a CEV person, in terms of the type and severity of individuals' underlying conditions; the manner in which people were added to the list; the time point that people were added to the list; and the extent to which people followed the shielding guidance.

⇒ Routinely collected healthcare data does not capture self-reported health and is likely to underestimate the burden of common mental disorders in the population.

## INTRODUCTION

In March 2020, Welsh Government and the Welsh National Health Service sought to protect people deemed clinically extremely vulnerable (CEV) to severe illness or death from COVID-19, advising them to 'shield' at home, that is, to remain indoors and minimise contact with others.[1] To identify CEV people, a shielded patient list (SPL) was created, using an algorithm based on clinical code lists and applied centrally to patients' electronic health records.[2] Additionally,

health professionals could add people to the list based on their clinical judgement. Shielding was in place from 23 March 2020 to 16 August 2020 and reintroduced from 22 December 2020 to 1 April 2021.[1] CEV children were encouraged to return to school at the end of August 2020 taking into account the low rate of severe disease and death from COVID-19 among children, balanced against the harms of a lack of schooling and socialisation.[3]

Studies have highlighted the detrimental impact of the COVID-19 pandemic on the mental health of CEV adults, with CEV individuals more likely to report increased depressive symptoms and anxiety[4] and to have a clinical record of anxiety or depression during the pandemic compared with those who were not CEV.[1] Meanwhile, studies have reported decreased diagnoses of mental health conditions in primary care across the population as a whole during the pandemic[5 6] (attributed to a reluctance to seek healthcare, or reduced access to services, rather than a decrease in need). However, there is limited evidence on how children accessed healthcare during the pandemic for their mental health, and no studies focusing on CEV children or children living with a CEV person.

Children are particularly vulnerable to indirect impacts of the pandemic.[7] Drawing on evidence from longitudinal surveys,[8–10] the department for education concluded that children's mental health declined during the pandemic, reporting that rates of probable mental health disorders were higher from 2020 than before.[11] The data also highlighted variation in mental health trajectories; children with long-term health conditions were more likely to experience poor mental health during the pandemic.[11] However, the extent to which these trends are attributable to the pandemic, or are a continuation of pre-existing upward trends, is unclear.

Almost 5000 CEV children were living in Wales by July 2020; approximately 3.9% of the Welsh CEV population.[12] Children with chronic illnesses are at increased risk of behavioural and emotional problems and psychiatric disorders compared with their peers,[13] but CEV children may be particularly susceptible to mental health difficulties relative to non-CEV children since the pandemic, due to additional restrictions imposed by shielding guidance, potentially exacerbating loneliness and isolation.[14] CEV children may have also experienced heightened health anxiety due to their potential higher risk of severe illness from COVID-19. Additionally, there were almost 14 400 school-aged children living with a CEV person in June 2020[15] who may be at greater risk of mental health difficulties due to both restrictions to protect the vulnerable members of their household, and fears of causing harm.[16]

We investigated the impact of the COVID-19 pandemic on use of healthcare services for anxiety or depression in Wales, for CEV children, children living with a CEV person and children in the general population, using routinely collected population-level linked data. The primary aim was to determine whether CEV children or children living with a CEV person were more likely to have a record for anxiety or depression in primary or secondary care during the pandemic compared with children in the general population. The secondary aim was to compare patterns of anxiety or depression in 2019/2020 and 2020/2021 between CEV children and the general population, to place the impact of COVID-19 and the shielding guidance in the context of longer-term patterns of healthcare use.

## METHODS

### Study design and data sources

This is a population-based cross-sectional cohort study using anonymised, linked, routinely collected health and administrative data for the population of Wales, UK, held in the Secure Anonymised Information Linkage (SAIL) Databank (www.saildatabank.com). Within the SAIL Databank, encrypted linkage fields are used to link data anonymously from various sources at individual and household level (online supplemental appendix pp 1–2); known as anonymised linking fields (ALFs) for individuals and residential anonymised linking fields (RALFs) for residences.[17 18] We used these to link multiple datasets in this study (table 1). General practices (GPs) opt-in to providing data to SAIL; currently, SAIL contains primary care data for around 80% of the Welsh population, and the available data are representative of the entire Welsh population with respect to age, sex and deprivation.[19] The SAIL Databank was interrogated using DB2 Structured Query Language.

### Patient and public involvement

The study protocol was presented at the SAIL consumer panel meeting prior to study commencement. This panel consists of members of the public with an interest in data and its uses to improve services and healthcare. The panel provided advice and feedback on the study design from a public perspective.

### Data access and cleaning methods

All authors had full access to all the data in the study. Data cleaning included deduplication and restructuring of the SPL prior to cohort creation and analysis and was undertaken by LEC.

### Study population and setting

We created three study cohorts for 2020: (1) CEV children (2) children living with at least one CEV person and (3) a general population group of children who were *not* identified as CEV or living with a CEV person, along with two further cohorts for 2019 for comparison purposes. Children who were both CEV and living with a CEV person were categorised as CEV. Figure 1 shows a flow diagram of the inclusion criteria for each cohort. We included all children aged 2–17 years who were alive, living in Wales and registered with a GP that supplies data to the SAIL Databank on 23 March 2020 and who had either an exact match on National Health Service (NHS)

**Table 1** Datasets used in this study

| Dataset | Description |
| --- | --- |
| Welsh Demographic Service dataset | A register containing demographic information about all individuals registered at a Welsh General Practice (GP) |
| COVID-19 shielded people list (CVSP) | A dataset containing information about clinically extremely vulnerable individuals in Wales, including reasons for shielding |
| Welsh Index of Multiple Deprivation 2019 | A dataset containing deprivation scores corresponding to all Lower-layer Super Output Areas (LSOAs; geographic units comprised of around 1600 individuals) in Wales[24] |
| Rural Urban Classification dataset | A dataset containing information on urban/rural categories corresponding to all LSOAs in Wales |
| Annual District Death Extract | A register containing details of all deaths of Welsh residents, including information regarding date and cause of death |
| Outpatient dataset | A dataset containing attendance information for all hospital outpatient appointments in Wales |
| National Community Child Health Database | A register of children born in Wales, containing data collected at birth, including a maternal anonymised linking field to link children with their biological mothers |
| Patient Episode Database for Wales | A dataset containing attendance and clinical information for all hospital admissions in Wales, including data regarding diagnoses |
| Care homes data | A dataset containing residential information about adult care homes in Wales |
| Welsh Longitudinal General Practice dataset | A dataset containing attendance and clinical information for all GP interactions including symptoms, diagnoses and prescriptions |

number or demographics (name, date of birth, gender code and postcode) or a probabilistic match of 90% or greater.[17] We excluded those for whom full demographic or residence data were not available.

### Cohort 1: CEV children

Cohort 1 consisted of all children who were identified as CEV (either by algorithm[2] or health professionals) between 23 March 2020 and 16 August 2020 (N=3769).

### Cohort 2: children living with at least one CEV person

To identify children living with a CEV person, we first identified the RALF for all CEV people in Wales as of 23 March 2020, including dates of residence. To minimise bias, we then adopted a conservative approach and included (1) children who were recorded as residents at the same RALF on 23 March 2020 and had an entry date of residence within 6 months of the entry date of the CEV person, (2) children born to mothers who were recorded as resident at the same RALF as the CEV person or (3) children who shared the same maternal ALF as a younger CEV child. We excluded (1) adult care homes as these were unlikely to contain children[20] and (2) RALFs containing more than 10 people, as the Unique Property Reference Number from which the RALF is derived is considered inaccurate in this case.[21] This resulted in a total of 20 033 children in cohort 2.

### Cohort 3: children in the general population

Cohort 3 consisted of all other children who were alive, living in Wales and registered with an SAIL-supplying GP on 23 March 2020 (N=415 009).

### CEV children in Wales in 2019 (pre-COVID-19 CEV children)

To explore longer-term patterns of anxiety or depression among children with the conditions included within the shielding guidance, we created a cohort of pre-COVID-19 'CEV' children who had similar health concerns to CEV children in a period prior to 2020. Creation of this prepandemic cohort enabled us to explore whether adverse mental health outcomes for CEV children during the pandemic were likely attributable to the shielding guidance, or whether CEV children experience poorer mental health compared with the general population regardless of having to shield. We included children who were alive, living in Wales and registered with an SAIL-supplying GP 1 year prior to the introduction of shielding (23 March 2019) and who had one or more of three of the health condition categories warranting inclusion on the SPL (respiratory illnesses, blood or bone cancer, and immunosuppression therapy). These categories were chosen as they only required the patient to have one of the listed codes within a given time period, and therefore children with these conditions could be identified with a high degree of certainty. We used International Classification of Diseases (ICD) V.10 diagnostic codes and Operating Procedures Codes Supplement (OPCS) Classification of Interventions and Procedures V.4 codes for these categories, which were taken from the SPL documentation and provided in the online supplemental appendix pp 4–5 . For comparison purposes, we also created a general population cohort of children aged 2–17 who were alive, living in Wales and registered with an SAIL-supplying GP as of 23 March 2019. A flow diagram of inclusion criteria for these cohorts is provided in the online supplemental appendix p 6. We performed a sensitivity analysis to confirm the validity of this approach (online supplemental appendix pp 7–9).

### Measures
#### Outcome of interest: risk of anxiety or depression

The outcome of interest was the first record of anxiety or depression in primary or secondary healthcare data during the COVID-19 pandemic (ie, 23 March 2020–31 January 2021, referred to as 2020/2021) and pre pandemic (23 March 2019–31 January 2020, referred to as 2019/2020).

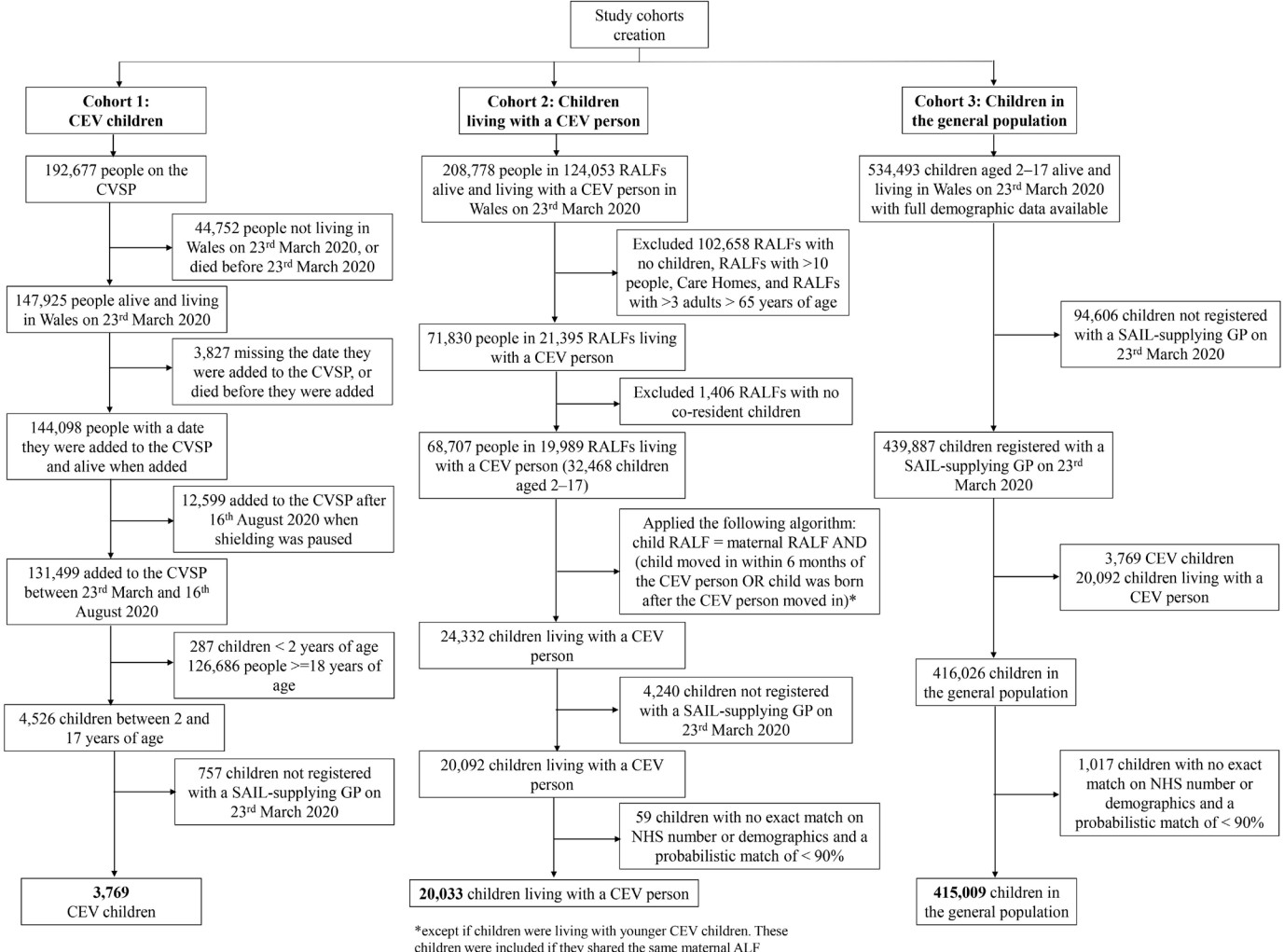

**Figure 1** Flow diagram of the inclusion criteria for the creation of three study cohorts: clinically extremely vulnerable children, children living with a clinically extremely vulnerable person and children in the general population. CEV, clinically extremely vulnerable; CVSP, COVID-19 shielded people list; GP, general practice; NHS, National Health Service; RALF, residential anonymised linking field; SAIL, Secure Anonymised Information Linkage.

We included any healthcare visit where anxiety or depression was documented in the electronic health record. We used validated Read V.2 codes to identify children with primary care records for anxiety[22] or depression[23] (including diagnoses, symptoms and prescriptions) in the Welsh Longitudinal General Practice dataset. Read codes are a hierarchical terminology system that encode clinical, diagnostic and therapeutic patient information and enable data entry of patient care information following a primary care consultation. Separate Read codes are used to record a patients' reported past medical history, or to note that a clinician is aware of a past condition. We used ICD-10 diagnostic codes to identify children with hospital admissions or outpatient appointments for anxiety or depression in the Patient Episode Database for Wales and outpatient dataset datasets. If children had multiple records of anxiety or depression during the relevant time periods, we sequenced these and selected the record with the earliest date. Code lists were reviewed by a clinician with expertise in child psychiatry (JT) and are provided

in the online supplemental appendix pp 10–14. The primary outcome was the *time to the first record* of anxiety or depression for the children in each cohort, and the secondary outcome was the *period prevalence* of anxiety or depression for the children in each cohort.

### Covariates: history of anxiety or depression
We used the same process to identify children in the study population with a 'recent' or 'past' history of anxiety or depression, defined as any record in the year prior to the pandemic (23 March 2019–23 March 2020), and any record occurring any time before 23 March 2019, respectively.[1]

### Covariates: demographics
We calculated age and determined Lower-layer Super Output Areas (LSOA) as of 23 March 2020. LSOA codes were derived from the Welsh Demographic Service Dataset based on the child's RALF, and used to ascertain deprivation quintiles and urban/rural classification by

linking to the Welsh Index of Multiple Deprivation 2019[24] and Rural Urban Classification datasets.

## Statistical analysis
We used R V.4.1.1 for statistical analyses. P values of <0.05 were considered statistically significant.

## Examining risk of anxiety or depression between the different cohorts in 2020/2021
We tested the hypothesis that there was no difference in the risk of having a record of anxiety or depression in 2020/2021 between the three cohorts (CEV children, children living with a CEV person and children in the general population). We plotted Kaplan-Meier survival curves for each cohort. We used Cox regression to calculate unadjusted and adjusted HRs with 95%CIs. We report three models examining the risk of having a record of anxiety or depression during the pandemic compared with the general population; (1) unadjusted, (2) adjusted for demographic factors (age group, sex, deprivation and rurality) and (3) adjusted for demographic factors and previous history of anxiety or depression (no history, recent history, past history or both recent and past history).

## Examining risk of anxiety or depression between the different cohorts in 2019/2020
We tested the hypothesis that there was no difference in the risk of having a record of anxiety or depression in 2019/2020 between two cohorts ('CEV' children and all other children living in Wales in 2019). As above, we calculated unadjusted and adjusted HRs, reporting three

models, and plotted Kaplan-Meier survival curves for each cohort.

## Comparing the risk of anxiety or depression in children between 2019/2020 and 2020/2021
We calculated the change in the period prevalence of anxiety or depression for CEV or 'CEV' children, and all other children living in Wales, between 2019 and 2020.

## Study reporting
This study is reported in accordance with the Reporting of Studies conducted using Observational Routinely-collected data guidelines[25] (online supplemental appendix pp 15–23).

## RESULTS
### Descriptive statistics and demographic characteristics of the study population
Demographic characteristics of the 2020/2021 study population are presented in table 2, and for the 2019/2020 study population in the online supplemental appendix p 24. For both years, there were greater proportions of boys, older children (aged 13–17), children living in the least and most deprived quintiles and children with a history of anxiety or depression in the CEV children than for the general population. In 2020/2021, a greater proportion of children living with a CEV person were older (aged 13–17) and had a history of anxiety or depression, compared with the general population

**Table 2** Characteristics of clinically extremely vulnerable (CEV) children, children living with a CEV person and a general population group of children neither CEV nor living with a CEV person, 2020/2021, Wales

| | | General population | CEV children | $\chi^2$ statistic, df, and p value, general population v. CEV children | Children living with a CEV person | $\chi^2$ statistic, df, and p value, general population v. children living with a CEV person |
|---|---|---|---|---|---|---|
| N | | 415 009 | 3769 | | 20 033 | |
| Sex (%) | Male | 212 311 (51.2) | 2184 (57.9) | $\chi^2$=68.6, df=1, p<0.001 | 10 247 (51.2) | $\chi^2$=0.0, df=1, p=0.989 |
| | Female | 202 698 (48.8) | 1585 (42.1) | | 9786 (48.8) | |
| Age group (years) (%) | 2–7 | 150 999 (36.4) | 1141 (30.3) | $\chi^2$=84.0, df=2, p<0.001 | 6692 (33.4) | $\chi^2$=93.7, df=2, p<0.001 |
| | 8–12 | 137 310 (33.1) | 1247 (33.1) | | 6678 (33.3) | |
| | 13–17 | 126 700 (30.5) | 1381 (36.6) | | 6663 (33.3) | |
| Deprivation quintile (Welsh Index of Multiple Deprivation 2019) (%) | 1 | 106 075 (25.6) | 1014 (26.9) | $\chi^2$=11.1, df=4, p=0.025 | 5058 (25.2) | $\chi^2$=10.8, df=4, p=0.029 |
| | 2 | 87 930 (21.2) | 773 (20.5) | | 4107 (20.5) | |
| | 3 | 73 106 (17.6) | 635 (16.8) | | 3611 (18.0) | |
| | 4 | 69 789 (16.8) | 589 (15.6) | | 3488 (17.4) | |
| | 5 | 78 109 (18.8) | 758 (20.1) | | 3769 (18.8) | |
| Rural/urban area (%) | Rural | 110 682 (26.7) | 971 (25.8) | $\chi^2$=1.5, df=1, p=0.217 | 5444 (27.2) | $\chi^2$=2.5, df=1, p=0.116 |
| | Urban | 304 327 (73.3) | 2798 (74.2) | | 14 589 (72.8) | |
| Any history of anxiety or depression (%) | Yes | 17 986 (4.3) | 368 (9.8) | $\chi^2$=261.5, df=1, p<0.001 | 1131 (5.6) | $\chi^2$=78.0, df=1, p<0.001 |
| | No | 397 023 (95.7) | 3401 (90.2) | | 18 902 (94.4) | |

**Table 3** Proportion of clinically extremely vulnerable children with different conditions contributing to underlying reasons to shield

| Reason for shielding | Number and percentage of clinically extremely vulnerable children (n=3769) |
|---|---|
| Rare diseases | 1227 (32.6%) |
| Organ disease | 735 (19.5%) |
| Respiratory illness | 660 (17.5%) |
| Immunosuppression therapy | 491 (13%) |
| Cancer | 147 (3.9%) |
| Transplant | 132 (3.5%) |
| Renal dialysis | 10 (0.3%) |
| General practice referred (*reason unknown*) | 582 (15.4%) |
| Other (*reason unknown*) | 22 (0.6%) |

In children whose reasons for shielding were known (3165/3769), 6.3% (198/3165) had more than 1 condition.

(table 2). The conditions leading to children being identified as CEV in 2020/2021 are shown in table 3.

### Risk of anxiety or depression in children in 2020/2021

Numbers of censored children in each group (due to death, migration or registration with a non-SAIL supplying GP) and numbers with a record for anxiety or depression are presented in table 4. Of those with a record, 5768/6251 (92.3%) presented to the primary care, while 483/6251 (7.7%) presented to the secondary care.

**Table 4** Number of children who were censored, with no event, or who had a record of anxiety or depression during 2020/2021

| | General population | Clinically extremely vulnerable children | Children living with a clinically extremely vulnerable person |
|---|---|---|---|
| N | 415 009 | 3769 | 20 033 |
| Died (%) | 12 (0.003) | 6 (0.2) | 0 (0.0) |
| Moved out of Wales (%) | 7297 (1.8) | 68 (1.8) | 245 (1.2) |
| Moved to a non Secure Anonymised Information Linkage-supplying general practice (%) | 5664 (1.4) | 31 (0.8) | 262 (1.3) |
| No event (%) | 396 267 (95.5) | 3505 (93.0) | 19 203 (95.9) |
| Anxiety or depression during the pandemic 2020/2021 (%) | 5769 (1.4) | 159 (4.2) | 323 (1.6) |

In the unadjusted model, both CEV children and children living with a CEV person were at significantly greater risk of having a record of anxiety or depression during the pandemic compared with the general population (HR=3.09, 95% CI=2.64 to 3.61, p<0.001 and HR=1.16, 95% CI=1.04 to 1.30, p<0.05, respectively). For CEV children, the HR remained significant when adjusting for demographic factors including age, sex, deprivation and rurality (table 5), and when adjusting for demographic factors and previous clinical history of anxiety or depression (table 6). However, for children living with a CEV person, the HR was no longer significant in either of the adjusted models. The unadjusted survival curves for each cohort are shown in figure 2.

### Risk of anxiety or depression in children in 2019/2020

In 2019/2020, in the unadjusted model there was an increased risk of having a record of anxiety or depression among the 'CEV' children compared with the general population (HR=1.94, 95% CI=1.31 to 2.87, p<0.001). This remained evident in the adjusted models (tables 7 and 8). The unadjusted survival curves for each cohort are shown in figure 3.

### Difference in the risk of records of anxiety or depression between 2019/2020 and 2020/2021

In 2019/2020, 'CEV' children had increased risk of having recorded anxiety or depression compared with children in the general population, and in 2020/2021 the risk ratio increased to 3.04 (table 9). This reflects a marked decline in presentation among children in the general population over this period (from 2.19% to 1.39%), alongside a small increase for CEV children (from 4.17% to 4.22%).

### DISCUSSION

In Wales, CEV children and children living with a CEV person were more likely to access health services for anxiety or depression during the pandemic than children in the general population. For CEV children, this pattern remained evident after adjusting for demographic differences and the likelihood of having a previous history of anxiety or depression. Although a small increase in risk was found for children living with a CEV person, after adjusting for demographic characteristics and previous history of anxiety and depression, this was no longer significant.

Both before and during the pandemic, the groups with the greatest risk of having a record for anxiety or depression were adolescents aged 13–17, and those with both a past and recent history of anxiety or depression. Females were also more likely than males to have a record for anxiety or depression in both time periods. These findings are in line with previous studies reporting worse mental health in female adolescents compared with male adolescents[26]; a higher prevalence of anxiety and depression in adolescents and females compared with younger children and males during the pandemic[27];

**Table 5** Multivariable analysis of risk factors for having a record of anxiety or depression during the COVID-19 pandemic (23 March 2020–31 January 2021), reported using HRs and 95% CIs (model adjusting for demographic factors only)

| | | HR | 95% CI | P value |
|---|---|---|---|---|
| Cohort | General population | Reference group | | |
| | Clinically extremely vulnerable (CEV) children | 2.81 | 2.40 to 3.29 | <0.001 |
| | Children living with a CEV person | 1.09 | 0.97 to 1.22 | 0.14 |
| Sex | Male | Reference group | | |
| | Female | 1.94 | 1.84 to 2.04 | <0.001 |
| Age group | 2–7 | Reference group | | |
| | 8–12 | 5.21 | 4.58 to 5.92 | <0.001 |
| | 13–17 | 19.39 | 17.20 to 21.86 | <0.001 |
| Deprivation quintile (Welsh Index of Multiple Deprivation 2019) | 1 (most deprived) | 1.13 | 1.05 to 1.22 | <0.01 |
| | 2 | 1.05 | 0.97 to 1.14 | 0.19 |
| | 3 | 1.05 | 0.97 to 1.14 | 0.23 |
| | 4 | 1.10 | 1.01 to 1.20 | <0.05 |
| | 5 (least deprived) | Reference group | | |
| Rural/urban area | Urban | Reference group | | |
| | Rural | 1.02 | 0.97 to 1.08 | 0.45 |

and worse mental health outcomes during the pandemic for those with pre-existing mental health difficulties.[28] This suggests that moving forward, it will be important to prioritise mental health support for female adolescents, and particularly for children who have concomitant physical and mental health conditions.

Given the detailed methodology used to identify CEV individuals in Wales,[2] we were able to develop a

**Table 6** Multivariable analysis of risk factors for having a record of anxiety or depression during the COVID-19 pandemic (23 March 2020–31 January 2021), reported using HRs and 95% CIs (model adjusting for both demographic and mental health factors)

| | | HR | 95% CI | P value |
|---|---|---|---|---|
| Cohort | General population | Reference group | | |
| | Clinically extremely vulnerable (CEV) children | 2.27 | 1.94 to 2.66 | <0.001 |
| | Children living with a CEV person | 1.02 | 0.91 to 1.14 | 0.746 |
| Sex | Male | Reference group | | |
| | Female | 1.58 | 1.50 to 1.66 | <0.001 |
| Age group | 2–7 | Reference group | | |
| | 8–12 | 4.56 | 4.01 to 5.18 | <0.001 |
| | 13–17 | 11.05 | 9.78 to 12.49 | <0.001 |
| Deprivation quintile (Welsh Index of Multiple Deprivation 2019) | 1 (most deprived) | 1.02 | 0.95 to 1.10 | 0.565 |
| | 2 | 0.98 | 0.90 to 1.06 | 0.614 |
| | 3 | 1.00 | 0.92 to 1.09 | 0.965 |
| | 4 | 1.07 | 0.98 to 1.16 | 0.127 |
| | 5 (least deprived) | Reference group | | |
| Rural/urban area | Urban | Reference group | | |
| | Rural | 1.02 | 0.96 to 1.02 | 0.464 |
| History of anxiety or depression | No history | Reference group | | |
| | Past history only | 5.13 | 4.75 to 5.53 | <0.001 |
| | Recent history only | 8.75 | 8.12 to 9.44 | <0.001 |
| | Both recent and past history | 18.98 | 17.52 to 20.55 | <0.001 |

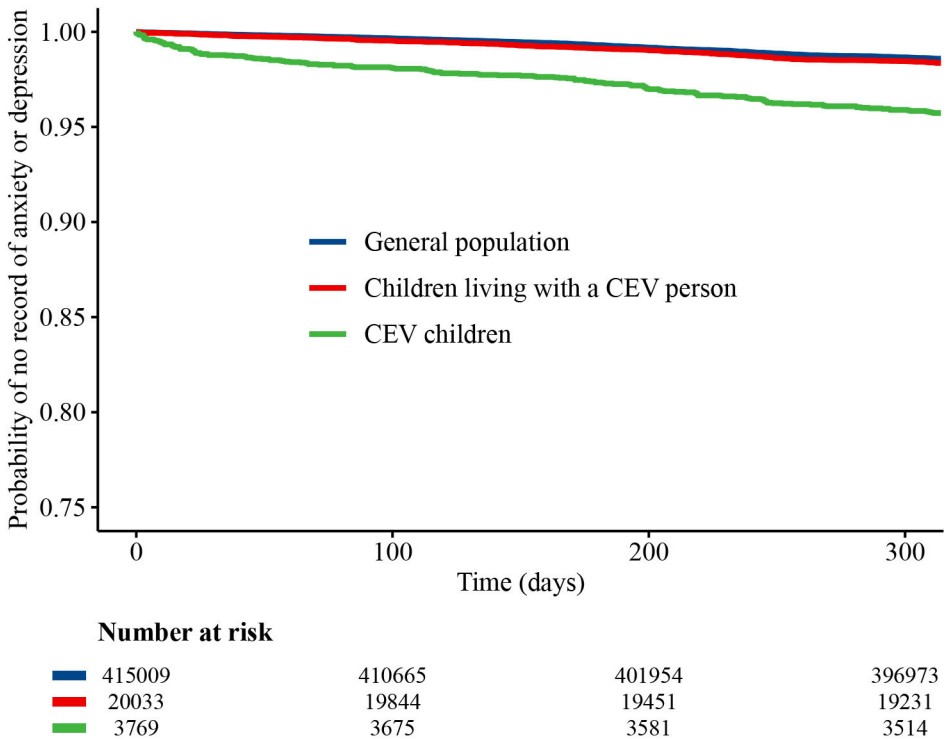

**Figure 2** Kaplan-Meier survival curves for each cohort, showing the time to first record of anxiety or depression between 23 March 2020 and 31 January 2021 (2020/2021). CEV, clinically extremely vulnerable.

comparable cohort of children with a subset of health conditions before the pandemic, using routine healthcare data. This enabled us to examine patterns of presentation for anxiety and depression among CEV children outside of the context of the pandemic. We found this group were at greater risk of having a record for anxiety or depression compared with children in the general population in 2019/2020, before COVID-19. In 2020/2021, CEV children remained at higher risk, and the difference was greater, although this is explained by a marked decline among children in the general population presenting to healthcare services with anxiety or depression during this time.

The reduction in presentations for anxiety and depression among children in the general population most likely reflects reduced access to NHS services during the pandemic. Other evidence suggests increased demand and unmet need for mental health support

**Table 7** Multivariable analysis of risk factors for having a record of anxiety or depression between 23 March 2019 and 31 January 2020, reported using HR and 95% CIs (model adjusting for demographic factors only)

| | | HR | 95% CI | P value |
|---|---|---|---|---|
| Cohort | General population | Reference group | | |
| | Clinically extremely vulnerable children | 2.03 | 1.37 to 3.01 | <0.001 |
| Sex | Male | Reference group | | |
| | Female | 1.85 | 1.77 to 1.93 | <0.001 |
| Age group | 2–7 | Reference group | | |
| | 8–12 | 4.60 | 4.17 to 5.08 | <0.001 |
| | 13–17 | 18.50 | 16.89 to 20.27 | <0.001 |
| Deprivation quintile (Welsh Index of Multiple Deprivation 2019) | 1 (most deprived) | 1.32 | 1.25 to 1.41 | <0.001 |
| | 2 | 1.22 | 1.15 to 1.31 | <0.001 |
| | 3 | 1.13 | 1.06 to 1.22 | <0.001 |
| | 4 | 1.09 | 1.02 to 1.17 | <0.05 |
| | 5 (least deprived) | Reference group | | |
| Rural/urban area | Urban | Reference group | | |
| | Rural | 0.97 | 0.93 to 1.02 | 0.21 |

**Table 8** Multivariable analysis of risk factors for having a record of anxiety or depression between 23 March 2019 and 31 January 2020, reported using HR and 95% CIs (model adjusting for both demographic and mental health factors)

| | | HR | 95% CI | P value |
|---|---|---|---|---|
| Cohort | General population | Reference group | | |
| | Clinically extremely vulnerable children | 2.03 | 1.37 to 3.01 | <0.001 |
| Sex | Male | Reference group | | |
| | Female | 1.54 | 1.48 to 1.61 | <0.001 |
| Age group | 2–7 | Reference group | | |
| | 8–12 | 4.13 | 3.74 to 4.56 | <0.001 |
| | 13–17 | 11.56 | 10.53 to 12.68 | <0.001 |
| Deprivation quintile (Welsh Index of Multiple Deprivation 2019) | 1 (most deprived) | 1.23 | 1.15 to 1.30 | <0.001 |
| | 2 | 1.14 | 1.07 to 1.22 | <0.001 |
| | 3 | 1.09 | 1.02 to 1.16 | <0.05 |
| | 4 | 1.05 | 0.98 to 1.13 | 0.17 |
| | 5 (least deprived) | Reference group | | |
| Rural/urban area | Urban | Reference group | | |
| | Rural | 0.97 | 0.92 to 1.02 | 0.19 |
| History of anxiety or depression | No history | Reference group | | |
| | Past history only | 4.50 | 4.21 to 4.81 | <0.001 |
| | Recent history only | 8.44 | 7.94 to 8.96 | <0.001 |
| | Both recent and past history | 16.97 | 15.85 to 18.18 | <0.001 |

in the UK, for children with and without pre-existing mental health problems, since 2020.[29][30] Findings from self-report surveys[8–10] and the current study suggest that the pandemic has widened the gap between need and access to mental healthcare for the general population of children in Wales, but additional data are required to unpack the relationship between self-reported mental health needs and presentation to healthcare services.

Meanwhile, the relatively stable period prevalence of anxiety and depression for CEV children in 2019/2020

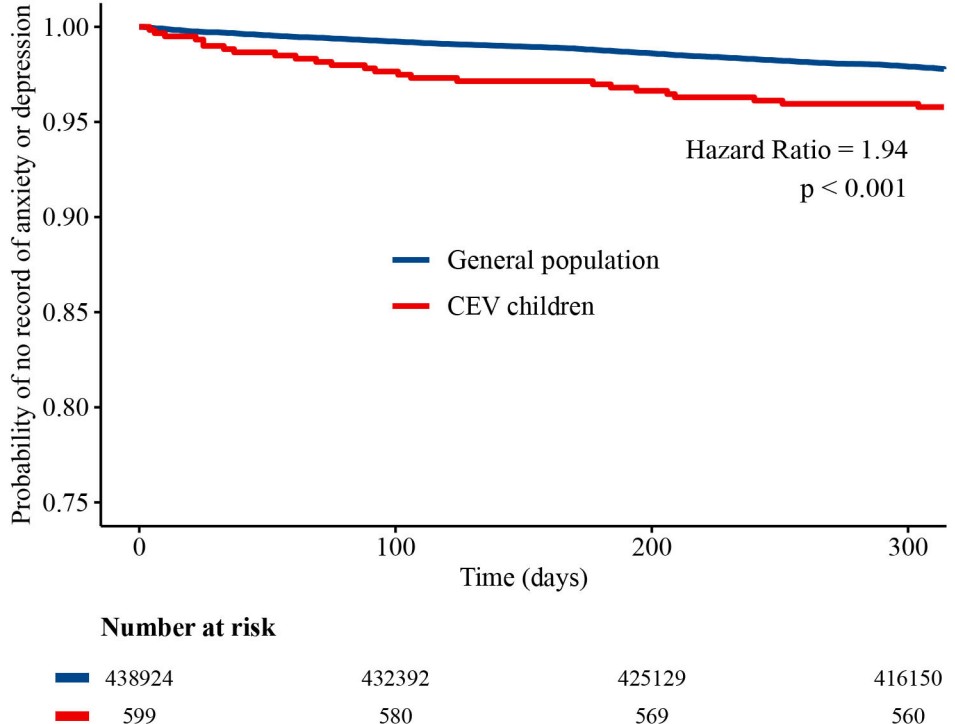

**Figure 3** Kaplan-Meier survival curves for each cohort, showing the time to first record of anxiety or depression between 23 March 2019 and 31 January 2020 (2019/2020). CEV, clinically extremely vulnerable.

**Table 9** Risk of records of anxiety or depression in children (aged 2–17 years) who were clinically extremely vulnerable (CEV) and those in the general population, in 2019/2020 and 2020/2021

| Time period | CEV children 2020/2021—children identified through shielded patient list 2019/2020—comparable cohort of children with the conditions listed in the shielded patient list | | | Children in the general population in Wales | | | |
| | No. children with recorded anxiety or depression | Total no. of children | Period prevalence (%) | No. children with recorded anxiety or depression | Total no. of children | Period prevalence (%) | Crude risk ratio |
| --- | --- | --- | --- | --- | --- | --- | --- |
| 2019/2020 (Pre-COVID-19) | 25 | 599 | 4.17 | 9620 | 438 924 | 2.19 | 1.90 |
| 2020/2021 (During COVID-19) | 159 | 3769 | 4.22 | 5769 | 415 009 | 1.39 | 3.04 |
| Percentage point change over time points | | | −0.05 | | | 0.8 | |

and 2020/2021 could indicate that this group did not experience an increase in mental health needs during the pandemic over and above past years, and that they had access to mental health support through existing care pathways for underlying conditions. Alternatively, if CEV children experienced the same increase in mental health needs as reported elsewhere among the general population, then these figures may mask unmet demand for mental health support among this group. A survey of adults supports the latter explanation, reporting increased anxiety and depressive symptoms among shielding individuals compared with non-shielding individuals.[4] However, we have found no UK surveys focusing on the mental health of CEV children.

Our finding that children living with a CEV person were at no greater risk of presenting with anxiety or depression during the pandemic compared with the general population (after controlling for other factors), could be interpreted in two ways. It is possible that the impact of the pandemic on the mental health of children living with a CEV person was not as great as for CEV children, but this seems unlikely given that research has suggested increased anxiety among children who were shielding their siblings.[16] Another explanation is that children living with a CEV person suffered from similar barriers to access to mental healthcare services to the general population and did not have the same routes to access that CEV children did. This explanation is supported by research with shielding families, which suggests that they have felt left behind and that children living with a CEV person may have 'fallen under the radar of educational and healthcare professionals'.[16]

### Strengths and limitations
To our knowledge, this is the first population-based study examining the effects of the COVID-19 pandemic on healthcare use for anxiety or depression among CEV children and children living with a CEV person in Wales. Linkage of population-based routinely collected data is a valuable method for generating evidence with a high level of external validity and applicability for policy-making. A strength of this study is the comparison of 2020/2021 data with prepandemic 2019/2020 data for CEV children and children in the general population. Another strength is the use of a novel approach using multiple linked datasets to identify a cohort of children living with a CEV person during the pandemic.

This study used the SPL to create cohorts of children identified as CEV and a cohort of children living with a CEV person. There was heterogeneity within the SPL in terms of the type and severity of individuals' underlying conditions; the manner in which people were added to the list (via the algorithm or clinical judgement); the time point that people were added to the list; and the extent to which people followed the shielding guidance. In addition, the impact of following shielding guidance is likely to have varied due to individual circumstances and the level of support received. The 2019 'CEV' cohort was a relatively small sample and for pragmatic reasons, only included children with a subset of the conditions included in the shielding guidance. To identify children living with a CEV person, we adopted a conservative approach and included children only if they were living with their mother. We took this approach in order to minimise bias and increase the generalisability of the findings; however, this approach is likely to have underestimated the number of children living with a CEV person. Finally, this study focused on healthcare use using clinical codes. Routinely collected healthcare data does not capture self-reported health, and is likely to underestimate the burden of common mental disorders in the population.[31] Focusing on healthcare use with routine data alone cannot tell us about the underlying reasons for changes in utilisation, or the scale of mental health need.

### Implications for policy and practice
Our findings have implications for recovery planning to prevent, mitigate and respond to the mental health impacts of the pandemic. We have shown changes in presentation to primary and secondary healthcare

services with anxiety and depression for CEV children and children in the general population during the pandemic, and there are concerns regarding potential increases in unmet mental health needs over time. As highlighted by UK organisations, such as the Centre for Mental Health,[32] services face challenges in tackling this demand. This has been recognised by Welsh Government, who invested an additional £9.4 million in children's mental health services in 2021.[33]

This novel linked data study contributes to our understanding of the direct and indirect impact of shielding on children's mental health in Wales during the COVID-19 pandemic. This evidence should be considered in light of additional, more detailed routine healthcare linkage studies, and national surveys, to provide a comprehensive understanding of the relationship between mental health support needs, expressed demands and care provision to better target services to those who need them the most.

Beyond the indirect impacts of the pandemic, our findings highlight the increased mental health needs of children with serious medical conditions. Given that these children are likely to have greater contact with healthcare services, signposting across services, including mental health services, is likely to be beneficial.

**Author affiliations**
[1]Population Data Science, Swansea University Medical School, Swansea, UK
[2]Research and Evaluation Division, Public Health Wales, Cardiff, UK
[3]Health Protection Division, Public Health Wales, Cardiff, UK
[4]Administrative Data Research Unit, Welsh Government, Cardiff, UK
[5]Department of Psychiatry, University of Oxford, Oxford, UK
[6]National Centre for Population Health and Wellbeing Research, Swansea University, Swansea, UK

**Acknowledgements** Adolescent Mental Health Data Platform (ADP) and the authors would like to acknowledge the data providers who supplied the datasets enabling this research study. The views expressed are entirely those of the authors and should not be assumed to be the same as those of ADP or MQ Mental Health Research Charity. This study makes use of anonymised data held in the Secure Anonymised Information Linkage (SAIL) Databank. We would like to acknowledge all the data providers who make anonymised data available for research.

**Contributors** LEC: study design, literature search, data curation, data analysis, figures, data interpretation, writing—original draft, writing—review and editing. KH: data analysis, data interpretation, supervision, writing—review and editing. JS: conceptualisation, study design, methodology, data curation, data analysis, data interpretation, supervision, writing—review and editing. TW: methodology, writing—review and editing. JT: validation, writing—review and editing. AJ: methodology, funding acquisition, writing—review and editing. AB: methodology, writing—review and editing. ARD: conceptualisation, study design, data analysis, data interpretation, supervision, funding acquisition, writing—review and editing. LEC and JS verified the underlying data. ARD was responsible for the overall content as the guarantor.

**Funding** This work was supported by the Adolescent Mental Health Data Platform (ADP). The ADP is funded by MQ Mental Health Research Charity (Grant Reference MQBF/3 ADP). This work was also supported by the National Centre for Population Health and Wellbeing Research, which is funded by Health and Care Research Wales. The funders had no role in the study design, data collection, data analysis, interpretation, writing of the report or the decision to submit the paper for publication.

**Competing interests** AJ is a member of the Welsh Government COVID-19 Technical Advisory Group and is also cochair of the Scientific Pandemic Insights Group on Behaviours, which is a subgroup of the Scientific Advisory Group for Emergencies advising the UK government. None of the other authors have any competing interests.

**Patient and public involvement** Patients and/or the public were involved in the design, or conduct, or reporting, or dissemination plans of this research. Refer to the Methods section for further details.

**Patient consent for publication** Not applicable.

**Ethics approval** This study used anonymised data and therefore did not require National Research Ethics Committee approval. Approval to access and link the data within the Secure Anonymised Information Linkage Databank was granted by the Information Governance Review Panel under project number 1265.

**Provenance and peer review** Not commissioned; externally peer reviewed.

**Data availability statement** Data may be obtained from a third party and are not publicly available. The data used in this study are available in the Secure Anonymised Information Linkage (SAIL) Databank at Swansea University (Swansea, UK) via the Adolescent Mental Health Data Platform, but, as restrictions apply, they are not publicly available. All proposals to use SAIL data are subject to review by an independent Information Governance Review Panel (IGRP). Before any data can be accessed, approval must be given by the IGRP. The IGRP carefully considers each project to ensure proper and appropriate use of SAIL data. When access has been granted, it is gained through a privacy-protecting safe haven and remote access system referred to as the SAIL Gateway. SAIL has established an application process to be followed by anyone who would like to access data via SAIL, details of which can be found at https://saildatabank.com/data/apply-to-work-with-the-data/.

**ORCID iDs**
Laura Elizabeth Cowley http://orcid.org/0000-0002-7757-4219
Ann John http://orcid.org/0000-0002-5657-6995
Amrita Bandyopadhyay http://orcid.org/0000-0003-2798-4030
Alisha R Davies http://orcid.org/0000-0002-8066-7264

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
