## [Reviewer comments · BMJ Open]

ARTICLE DETAILS

TITLE (PROVISIONAL)	Effects of the COVID-19 pandemic on the mental health of clinically extremely vulnerable children and children living with clinically extremely vulnerable people in Wales: a data linkage study
AUTHORS	Cowley, Laura; Hodgson, Karen; Song, Jiao; Whiffen, Tony; Tan, Jacinta; John, Ann; Bandyopadhyay, Amrita; Davies, Alisha

VERSION 1 – REVIEW

REVIEWER	Eapen, Valsamma South Western Sydney Local Health District, ICAMHS
REVIEW RETURNED	30-Oct-2022

GENERAL COMMENTS	This important data linkage study provides a nationwide analysis on the impact of the COVID-19 pandemic on the mental health care needs among clinically extremely vulnerable (CEV) children or children living with a CEV person. Methods section: 1) Subgroup analysis – It is understood that the authors used a conservative approach. However, it would be useful to explore, using subgroup analysis, whether the COVID-19 pandemic had similar impact on children shielded for unknown reasons compared to those shielded for known reasons. Results and discussion: The authors have reported on unadjusted and adjusted hazard ratios for the first record of anxiety or depression in primary or secondary healthcare data during the COVID-19 pandemic and pre-pandemic periods. However, in terms of clinical implications, it is important to understand the sociodemographic factors associated with the above outcome. Whilst this has been tabulated in the appendix section, it is recommended that this be added in the results section along with a brief discussion with plausible explanation in the discussion section. Figures: Figures need to have higher resolution for better readability. Appendix section: Some of the information in the appendix section could be concise and be better cited as reports and added to the reference section.
---

REVIEWER	Smith, Melissa The University of Alabama at Birmingham, Biostatistics
REVIEW RETURNED	08-Mar-2023

GENERAL COMMENTS	Peer review of “Effects of the COVID-19 pandemic on the mental health of clinically extremely vulnerable children and children living with clinically extremely vulnerable people in Wales: A data linkage study” In this study, the authors evaluate whether there is a greater risk of records of anxiety and depression in CEV children and children living with a person who is considered CEV compared to children in the general population in Wales during the COVID-19 pandemic. They also look at records of anxiety and depression prior to the onset of the COVID-19 pandemic. I appreciate that the authors leverage numerous large national databases along with record linkage techniques to create and evaluate these different cohorts. Their study is quite interesting. However, I have several comments and points for clarification that should be addressed to be considered for publication. My primary comments relate to the outcome definition. Please note that my comments specifically pertain to the study design and statistical analyses in this manuscript, as this is my area of expertise. Comments:  • Outcome variable: the authors should provide additional clarity and justification on their outcome variable.  o From my understanding of the outcome definition, the outcome variable seems to be a blend of incidence and prevalence. I suggest the authors focus on either incidence or prevalence. The current outcome is not quite anxiety or depression incidence, because it does not always document a new diagnosis. The authors should not refer to this as “cumulative incidence” or “risk of anxiety/ depression” without additional verbiage such as “risk of an anxiety/depression record within the time period of interest”. Why not consider estimating anxiety/ depression period prevalence for each cohort instead? This seems to align more closely with how the data were collected. o It was unclear to me whether the authors were specifically looking at healthcare visits for mental health conditions or whether another visit, such as an annual checkup, where anxiety was listed in the patient’s chart under health history, was included in this outcome. Please add additional details to clarify the outcome definition. o I am also wondering how the authors selected “first record of anxiety or depression in primary or secondary healthcare data during the COVID-19 pandemic and pre-pandemic”? Why is time-to-first record of anxiety or depression of interest? Is the timing from March 23rd both in 2019 and in 2020 a clinically important outcome for patients? Is this clinically relevant for the group of children who have been diagnosed with anxiety/depression in the past? • Survival analysis methods:  o The authors use survival analysis techniques such as Kaplan-Meier curves and Cox regression models, but they did not provide justification for why the timing is important. If prevalence, or risk of current anxiety/depression, is of interest, different statistical methods should be used to analyze whether the probability of having current anxiety/depression is different in CEV
--

	children and children in the general population at the two time points.  o In the Kaplan-Meier plots, “survival probability” should be changed to reflect the probability being plotted in this study. o In Figure 2, what was “time 0” for children who were identified as CEV after March 23rd? Was “time 0” March 23rd for all CEV children or did it vary based on when children were identified as CEV? • Presentation and interpretation of results:  o The authors should put more focus on interpreting the results that include past history of anxiety/ depression than on the unadjusted analysis, since this had a very large effect on risk of first record of depression (HR = 18.98 when comparing the risk of record for CEV children with both a recent and past history of anxiety/depression to children with no history). This large effect of past anxiety/depression should also be discussed in the manuscript. o It would be helpful to combine Tables 5 and 6 to facilitate the comparison of hazard ratios pre and post the COVID-19 pandemic. • Additional comments:  o Were there children who fell into both the CEV cohort and the children living with CEV cohort? Please describe how these children were categorized. o Page 9, line 1: Typo: “ALF” should be “RALF” o Page 9, line 5: Please add a sentence about why children in care homes and in RALFS containing more than 10 people were excluded from Cohort 2. Also, how did you define care homes?
--	--

VERSION 1 – AUTHOR RESPONSE

Reviewer 1:

1. Methods section: 1) Subgroup analysis – It is understood that the authors used a conservative approach. However, it would be useful to explore, using subgroup analysis, whether the COVID-19 pandemic had similar impact on children shielded for unknown reasons compared to those shielded for known reasons.

Our response: We understand the reviewers’ interest in this question, however the “shielding reason unknown” category reflects the way in which the data was collected. This category accounts for 15.4% of shielding children (Table 3) and is where a GP identified a child as vulnerable to COVID-19 and added them to the “shielding list”, but did not record a specific reason for shielding. The family and child may understand the vulnerability but this is not identified in the GP records, and this group is likely to be a very heterogeneous group. The aim of the study was to explore the mental health of all children who were identified as having to shield, irrespective of their underlying conditions, because this reflected the situation at that time during the pandemic when there was no distinction made between these groups.

2. Results and discussion: The authors have reported on unadjusted and adjusted hazard ratios for the first record of anxiety or depression in primary or secondary healthcare data during the COVID-19 pandemic and pre-pandemic periods. However, in terms of clinical implications, it is important to understand the sociodemographic factors associated with the above outcome. Whilst this has been tabulated in the appendix section, it is recommended that this be added in the results section along with a brief discussion with plausible explanation in the discussion section.

Our response: We agree that it would be useful to include the full model results in the main manuscript. We have now replaced the original Table 5 in the manuscript with the two tables on

pages 25 and 26 of the original Appendix document, which show the full results for the two adjusted models for the pandemic period, and are now labelled Tables 5 and 6 in the revised manuscript. Similarly, we have replaced the original Table 6 in the manuscript with the two tables on pages 27 and 28 of the original Appendix document, which show the full results for the two adjusted models for the pre-pandemic period. These tables are now labelled Tables 7 and 8 in the revised manuscript. We have also added a paragraph in the discussion relating to the sociodemographic factors, as follows: *“Both before and during the pandemic, the groups with the greatest risk of having a record for anxiety or depression were adolescents aged 13–17, and those with both a past and recent history of anxiety or depression. Females were also more likely than males to have a record for anxiety or depression in both time periods. These findings are in line with previous studies reporting worse mental health in female adolescents compared to males²⁶; a higher prevalence of anxiety and depression in adolescents and females compared to younger children and males during the pandemic²⁷; and worse mental health outcomes during the pandemic for those with pre-existing mental health difficulties²⁸. This suggests that moving forward, it will be important to prioritise mental health support for female adolescents, and particularly for children who have concomitant physical and mental health conditions.”* We have now also updated the reference list and the references in the text to account for these three new citations.

3. Figures: Figures need to have higher resolution for better readability.

Our response: We have now enhanced the quality of the figures and resubmitted them as PDF files. We will work with the editors if any further improvements are needed.

4. Appendix section: Some of the information in the appendix section could be concise and be better cited as reports and added to the reference section.

Our response: The Appendix section contains important additional information about the SAIL databank, the ethical approvals process, the code lists that we have used, the creation of the pre-pandemic study cohorts, the sensitivity analysis that we undertook, and the required reporting checklist. We have provided a separate reference section within the Appendix document of key references to the SAIL Databank information governance procedures. We have also now removed four of the supplementary tables into the main manuscript as per your previous suggestion. We do not have a platform to publish the information in the Appendix document as separate reports, and it is important that the additional information is kept linked to the main manuscript.

Reviewer 2:

1. Outcome variable: the authors should provide additional clarity and justification on their outcome variable. From my understanding of the outcome definition, the outcome variable seems to be a blend of incidence and prevalence. I suggest the authors focus on either incidence or prevalence. The current outcome is not quite anxiety or depression incidence, because it does not always document a new diagnosis. The authors should not refer to this as “cumulative incidence” or “risk of anxiety/ depression” without additional verbiage such as “risk of an anxiety/depression record within the time period of interest”. Why not consider estimating anxiety/ depression period prevalence for each cohort instead? This seems to align more closely with how the data were collected.

Our response: Thank you for pointing this out, we agree that we have not measured cumulative incidence, but rather period prevalence (the proportion of children with anxiety/depression during the two given time periods). We have updated all instances of “cumulative incidence” in the text and Table 9 with “period prevalence”. In addition, we have checked that where we refer to “risk of an anxiety/depression record” we have also referred to the relevant time period.

2. It was unclear to me whether the authors were specifically looking at healthcare visits for mental health conditions or whether another visit, such as an annual checkup, where anxiety was listed in the patient’s chart under health history, was included in this outcome. Please add additional details to clarify the outcome definition.

Our response: We included any healthcare visit where a Read code or ICD-10 code for anxiety or depression was recorded. There are separate Read codes used to record the patient's reported past medical history, or to note that a clinician is aware of the past condition. We have now added additional detail to the "Outcome of interest: Anxiety or depression" section, to clarify what healthcare visits were included, explain what Read codes are used for and explain that separate Read codes exist for recording past health history.

3. I am also wondering how the authors selected "first record of anxiety or depression in primary or secondary healthcare data during the COVID-19 pandemic and pre-pandemic"? Why is time-to-first record of anxiety or depression of interest? Is the timing from March 23rd both in 2019 and in 2020 a clinically important outcome for patients? Is this clinically relevant for the group of children who have been diagnosed with anxiety/depression in the past?

Our response: We selected the first record of anxiety/depression in primary or secondary healthcare data by sequencing the dates that a Read code or ICD-10 code for anxiety or depression was recorded for each child within the relevant time period, using inpatient (Patient Episode Database Wales), primary care (Welsh Longitudinal General Practice Dataset) and outpatient datasets, and then selecting the earliest date. We have now added a sentence to the methods to clarify this: *"If children had multiple records of anxiety or depression during the relevant time periods, we sequenced these and selected the record with the earliest date."* With regards to time-to first-record, the timing from 23rd March 2020 is clinically relevant because we were interested in looking at what time point following the introduction of the shielding guidance were children at risk of first presentation with anxiety or depression, to inform action in case of future waves or future pandemics. As can be seen in the Kaplan Meier plots, there was a gradual accumulation of presentations for anxiety/depression over time, suggesting an increase in mental health difficulties as the pandemic continued and the shielding guidance remained in place. For the 2019 cohort, the timing was less clinically relevant, however this cohort and analysis was constructed in order to assess and compare mental health in an equivalent time period, in a group of "CEV" children prior to the pandemic, in the absence of any shielding guidance. In terms of children who were diagnosed with anxiety/depression in the past, as outlined in the methods, we identified the children in each cohort with either a recent or past history of anxiety or depression, or both, and included these as covariates within our models. This enabled us to assess whether children with a previous history of anxiety or depression were more likely to go on to have a record of anxiety/depression during the pandemic, which we found was indeed the case. We have now included the full models in the main manuscript and a paragraph in the discussion about the risk factors, including previous history of anxiety/depression.

4. Survival analysis methods: The authors use survival analysis techniques such as Kaplan-Meier curves and Cox regression models, but they did not provide justification for why the timing is important. If prevalence, or risk of current anxiety/depression, is of interest, different statistical methods should be used to analyze whether the probability of having current anxiety/depression is different in CEV children and children in the general population at the two time points.

Our response: Please see the response to the above comment on why timing is important. The hazard ratio (HR) estimated by a Cox regression model, estimates the difference in risk of an outcome of interest between different groups and is therefore an appropriate statistical test to answer the specified research questions. Other statistical methods, such as logistic regression or generalized linear models (e.g. Poisson regression) may also be appropriate. We chose Cox regression over logistic regression because we wanted to capture first presentation of anxiety/depression and not just whether anxiety/depression had occurred. In addition, we chose Cox regression instead of Poisson regression because Poisson models assume a constant hazard over time, while Cox regression models do not. We couldn't assume that the chance of developing anxiety/depression during the pandemic was the same over time, as the introduction of the shielding guidance was a time point when families were required to drastically change their way of life.

5. In the Kaplan-Meier plots, "survival probability" should be changed to reflect the probability being plotted in this study.

Our response: We have now changed the y-axis labels on the Kaplan-Meier plots to read "Probability of no record of anxiety or depression".

6. In Figure 2, what was “time 0” for children who were identified as CEV after March 23rd? Was “time 0” March 23rd for all CEV children or did it vary based on when children were identified as CEV?

Our response: Time 0 was 23rd March for all children.

7. Presentation and interpretation of results: The authors should put more focus on interpreting the results that include past history of anxiety/ depression than on the unadjusted analysis, since this had a very large effect on risk of first record of depression (HR = 18.98 when comparing the risk of record for CEV children with both a recent and past history of anxiety/depression to children with no history). This large effect of past anxiety/depression should also be discussed in the manuscript.

Our response: We agree, we have now included the full modelling results in the main manuscript as also suggested by Reviewer 1, and we have now also added in a paragraph to the discussion, see our response to Reviewer 1 comment number 2.

8. It would be helpful to combine Tables 5 and 6 to facilitate the comparison of hazard ratios pre and post the COVID-19 pandemic.

Our response: We have now replaced Tables 5 and 6 with the full, adjusted modelling results that were previously included in the Appendix, as requested by Reviewer 1 (comment 2). This means that the results for during the pandemic are now presented in the new Tables 5 and 6, and the results for pre-pandemic are presented in the new Tables 7 and 8.

9. Additional comments: Were there children who fell into both the CEV cohort and the children living with CEV cohort? Please describe how these children were categorized.

Our response: Yes, there were children who fell into both cohorts, these children were categorized as CEV. We have added a sentence to clarify this in the “Study population and setting” section.

10. Page 9, line 1: Typo: “ALF” should be “RALF”

Our response: ALF is the correct term here. A maternal ALF enables mothers to be linked to their biological children. For this step of the cohort creation, we wanted to ensure we were picking up siblings who were living with younger CEV children, as these children were not captured in the previous steps.

11. Page 9, line 5: Please add a sentence about why children in care homes and in RALFS containing more than 10 people were excluded from Cohort 2. Also, how did you define care homes?

Our response: Care homes refer to adult care homes only. These were excluded because it was unlikely that any children would be living there. Care homes were identified using the Care Homes Data within SAIL, which contains residential information about known adult care homes in Wales (this is described in Table 1, but we have now made it clear that care homes refer to adult care homes only). RALFs containing more than 10 people were excluded because the Unique Property Reference number (UPRN) on which a RALF is based, is known to be inaccurate in this case and we could not be sure that people identified as living in these RALFs were truly living together. We have now added detail to explain why care homes and RALFs containing more than 10 people were excluded from Cohort 2.

VERSION 2 – REVIEW

REVIEWER	Smith, Melissa The University of Alabama at Birmingham, Biostatistics
REVIEW RETURNED	03-May-2023
GENERAL COMMENTS	The authors have addressed my comments and I recommend the manuscript is accepted after a small revision.

	The additional revision I suggest is related to more clearly describing the two summary measures of risk of anxiety/depression when the outcome is first introduced in the paper. The authors have appropriately modified the results to refer to their outcome as period prevalence or risk of presenting to a healthcare visit with anxiety/ depression within the time period under study. However, under the subsection "Outcome of Interest: Anxiety or depression" within the "Measures" section, period prevalence and time to first record of anxiety/ depression should be clearly mentioned here as the two ways they will summarize anxiety/ depression risk across cohorts. "Outcome of Interest: Anxiety or depression" is vague and should be changed to something like "Outcome of Interest: Period prevalence of anxiety or depression." Was period prevalence the primary outcome and timing the secondary outcome? I think laying these out more clearly at the start would be helpful rather than mentioning each summary measure later in the text. For example, including something like "Using the first records of anxiety/ depression recorded for the children in each cohort, our primary outcome was summarized as..." and "our secondary outcome was summarized as..." Otherwise, the manuscript looks great, and the revisions strengthen the paper.
--	--

VERSION 2 – AUTHOR RESPONSE

Reviewer 2:

1. The authors have addressed my comments and I recommend the manuscript is accepted after a small revision.

Our response: Thank you, we have made the small revision you have suggested.

2. The additional revision I suggest is related to more clearly describing the two summary measures of **risk of anxiety/depression** when the outcome is first introduced in the paper. The authors have appropriately modified the results to refer to their outcome as **period prevalence or risk of presenting to a healthcare visit with anxiety/ depression within the time period under study**. However, under the subsection "Outcome of Interest: Anxiety or depression" within the "Measures" section, **period prevalence and time to first record of anxiety/ depression should be clearly mentioned here as the two ways they will summarize anxiety/ depression risk across cohorts**. "Outcome of Interest: Anxiety or depression" is vague and should be changed to something like "Outcome of Interest: Period prevalence of anxiety or depression."

Was period prevalence the primary outcome and timing the secondary outcome? I think laying these out more clearly at the start would be helpful rather than mentioning each summary measure later in the text. For example, including something like "Using the first records of anxiety/ depression recorded for the children in each cohort, our primary outcome was summarized as..." and "our secondary outcome was summarized as..."

Our response: We have now changed the heading of the subsection from simply "Outcome of Interest: Anxiety or depression" to "Outcome of Interest: Risk of anxiety or depression". We have also added the following text to the end of the subsection: "The primary outcome was the *time to the first*

record of anxiety or depression for the children in each cohort, and the secondary outcome was the *period prevalence* of anxiety or depression for the children in each cohort.” We have also added a new subheading to the Statistical analysis section, to make the distinction between the survival analyses and the period prevalence calculations even clearer.

3. Otherwise, the manuscript looks great, and the revisions strengthen the paper.

Our response: Thank you for your positive comments.